

# *In situ* mechanical behavior of mineralized collagen fibrils in murine cortical bone is altered by aging and disuse

Fan Li[1,2,*], Fa Liu[2,*], Chenxi Ren[2], Shuyang Zhang[2], Zhe Wang[2] and Pengfei Yang[2]

[1] Hospital of Northwestern Polytechnical University, Xi'an, Shaanxi, China
[2] Key Laboratory for Space Bioscience and Biotechnology, Engineering Research Center of Chinese Ministry of Education for Biological Diagnosis, Treatment and Protection Technology and Equipment, School of Life Sciences, Northwestern Polytechnical University, Xi'an, Shaanxi, China
* These authors contributed equally to this work.

Corresponding author
Pengfei Yang, yangpf@nwpu.edu.cn

## ABSTRACT

Mineralized collagen fibrils (MCFs) are the fundamental building blocks of bone, determining its mechanical properties. Aging and disuse are known to impair bone mechanics, but their specific effects on the nanoscale, *in situ* mechanical behavior of MCFs remain poorly understood. The present study utilized a murine model involving adult and aged mice, with a subset from each age group subjected to disuse through hindlimb unloading ($n = 6$). To investigate the *in situ* nanomechanical response of bone, murine tibiae were tensile-loaded within a custom-made axial loading device while being simultaneously scanned with atomic force microscopy (AFM). The bone surface was partially demineralized to expose the collagen fibrils. High-resolution AFM imaging in tapping-mode was then employed to quantitatively assess the morphological changes and nanomechanical properties of MCFs throughout the bone's elastic deformation process. In the adult murine tibia, the initial response to load was characterized by fibril reorientation and an increase in the *in situ* elastic modulus of MCFs, indicating stretching. It was followed by a sliding phase between adjacent fibrils. In contrast, the aged bone exhibited fibril sliding at the onset of loading, accompanied by a gradual decrease in the elastic modulus of MCFs. The nanomechanical alterations induced by disuse were more pronounced in aged mice compared to adults. The present findings demonstrate that aging and disuse significantly alter the nanoscale deformation mechanisms of bone, shifting the response from fibril stretching to predominant sliding. It provides novel evidence for a unique, age-dependent deformation mechanism at the fibrillar level, enhancing the current understanding of how aging and disuse impair bone quality and mechanics.

## INTRODUCTION

Collagen fibrils are the most abundant and widely used protein polymers in animals. They are the structural basis for tissue and organ formation, and are primarily found in the extracellular matrix (*Holmes et al., 2018*). Their tissue-specific layered structure is the key
toughness element of composite materials. At the microscopic scale, collagen fibrils combine with hydroxyapatite (HA) mineral crystals to form mineralized collagen fibrils (MCFs). HA is deposited directionally in the gap regions of adjacent fibers or outside the fibrils. Mineral deposition can enhance the mechanical properties of tissues (*Silver, Jaffe & Shah, 2018*; *Garnero, 2015*). A unique and interesting feature of collagen fibers is that collagen molecules are filled in a quarter staggered manner to form D-band periodicity, a pattern of repeated bands of approximately 67 nm with periodic horizontal stripes alternating between light and dark (*Kontomaris, Stylianou & Malamou, 2022*). Collagen fibers are millimeters in length and range in diameter from a few nanometers to ~500 nm, depending on the tissue and developmental stage (*Ushiki, 2002*). As a basic component of many tissues and organs, collagen fibers play a vital role in many conditions (*Garnero, 2015*).

Collagen fibrils primarily serve a biological function through their mechanical properties (*Harley et al., 1977*). Over the past few decades, these properties have been extensively investigated using diverse techniques, including Brillouin spectroscopy (*Harley et al., 1977*; *Cusack & Miller, 1979*), force spectroscopy (*van der Rijt et al., 2006*), X-ray diffraction (*Sasaki & Odajima, 1996*), microelectromechanical systems (*Reffay et al., 2011*; *Shen et al., 2008*; *Eppell et al., 2006*), steered molecular dynamics simulations of tropocollagen-like molecules (*Lorenzo & Caffarena, 2005*), and AFM-based nanoindentation (*Stylianou et al., 2019*; *Hang & Barber, 2011*; *Svensson et al., 2010*). Synchrotron X-ray scattering studies have demonstrated that *in situ* tensile testing can directly quantify deformation mechanisms at the nanoscale (*Gupta et al., 2005*). Bone deformation arises from a combination of tensile deformation within collagen fibrils and shear deformation in the interfibrillar matrix, resulting in non-uniform strain distribution (*Gupta et al., 2005*). Uniaxial tensile testing, longitudinal strain analysis, and radial nanoindentation experiments reveal that type I collagen fibers show non-linear mechanical behavior with initial strain stiffening (0–15% strain) followed by strain softening (15–25% strain) (*Gupta et al., 2006*). *In situ* AFM-nanoindentation tensile tests further indicate that MCFs undergo early-stage elongation, characterized by increased D-periodic spacing, followed by fibril reorientation during later deformation phases (*Yang et al., 2018a*). Notably, the radial elastic modulus of collagen fibrils remains constant under tested loading conditions (*Yang et al., 2018a*). The mechanical properties of collagen fibers are crucial to cell-matrix interactions and are largely dictated by their molecular architecture (*Stylianou et al., 2019*). Understanding these properties is essential for elucidating tissue biophysics as well as for deciphering the microstructural organization of collagen fibrils themselves (*Stylianou et al., 2019*). During tendon stretching, low-strain regimes induce tendon flattening owing to gradual force application, followed by straightening of the molecular kinks in collagen fibrils. At higher strains, molecular sliding within the fibrillar structure becomes predominant. The quality of bone tissue deteriorates with age because of progressive alterations in bone mineral composition, collagen structure and cross-linking patterns, water content, and non-collagenous protein organization (*Burr, 2019*). The material properties of collagen fibrils are significantly influenced by two key factors: their mineralization state and the accumulation of advanced glycation end-products (AGEs)

(*Bonicelli et al., 2022*). Age-related experimental studies on bone tissue have further demonstrated a positive correlation between aging and increased mineral content (*Bonicelli et al., 2022*). Furthermore, elevated mineralization levels have been shown to enhance the elastic modulus of MCFs while simultaneously promoting brittle fracture behavior (*Zioupos, Kirchner & Peterlik, 2020*; *Liu et al., 2014*). Changes in mineral content caused by aging affect the elasticity, post-yield, and final properties of cortical bone. Disuse, a common mechanical environment in the aging stage, aggravates the declining mechanical properties of aging bones (*Lin, Dimitriadis & Horkay, 2007*). Excessive mineralization of collagen associated with aging may result in bone tissue stiffness and impaired bone remodeling (*Wallace, 2015*). Both enzyme-mediated and non-enzyme-mediated collagen cross-linking increases with age (*Sizeland et al., 2020*). Accumulation of non-enzyme cross-linking AGEs caused by aging may diminish interfibrillar sliding capacity, reduce energy dissipation, induce accumulation of microdamage, affect osteoblast activity, decrease bone properties, and affect charge disposition, thus altering fibril structure (*Chen et al., 2021*).

Most experiments have focused on the mechanical response of bone materials through multiple techniques. The *in situ* characterization of bone mechanical changes at nanoscale levels using advanced methodologies remains a technical challenge persistently addressed by researchers. AFM nanoindentation has emerged as a leading technology enabling nanoscale characterization of biological samples and is particularly effective for evaluating the nano-mechanics of individual collagen fibrils (*Hang & Barber, 2011*). In the present study, the deformation mechanisms of MCFs in adult *vs.* aged murine models under mechanical loading were systematically investigated. After controlled demineralization, the MCF morphology in cortical bone was dynamically monitored using AFM during deformation events. The *in situ* mechanical properties were quantitatively assessed using AFM-based nanoindentation. Eventually, the nanoscale deformation mechanisms of bone and the functional contributions of MCFs to the macroscopic mechanical behavior of bone were comprehensively discussed.

## MATERIALS AND METHODS

### Samples

#### *Animal care and housing*

All experimental protocols were approved by the Animal Ethics Committee of the Northwestern Polytechnical University (No. 2018036) and conducted in accordance with local regulations. Twenty-four male BALB/c mice (4 weeks old) were purchased from the Lab Animal Centre of Xi'an Jiaotong University (Xi'an, Shaanxi, China). Mice were housed in cages under controlled environmental conditions (temperature: 23 ± 1 °C, humidity: 50 ± 10%, 12-h light/dark cycle). Animals were group-housed at a density of four animals per cage to allow social interaction. Standard laboratory chow and water were provided *ad libitum*. Environmental enrichment included nesting materials and tunnels, which were regularly rotated to minimize stress.

## Experimental intervention of murine hindlimb unloading

Mice were reared to reach the sampling age of 8 months (adult group) and 18 months (aged group), with twelve mice for each age group (Fig. 1). Sample size calculations were performed using preliminary data and published effect sizes from similar experimental paradigms to ensure sufficient statistical power (*Liu et al., 2022*). Mice of each age group were randomly divided into the control group and hindlimb unloading (HLU) group, with six mice per group; the personnel conducting the behavioral tests and data analysis were unaware of the group allocations.

The tail was cleaned with 70% ethanol and air-dried. A lightweight surgical tape was attached to the proximal third of the tail, avoiding obstruction of blood flow or nerve function. The harness was connected to a swivel pulley system mounted atop the cage, allowing free 360° rotation and unrestricted forelimb movement. Mice were suspended at a 30° head-down tilt, ensuring that the hindlimbs remained elevated without contact with the cage surface for 28 days (*De Souza et al., 2005*). This position mimics musculoskeletal unloading. No obvious weight loss (>20%), self-mutilation, or prolonged lethargy of mice were observed during the experiment. At the study endpoint, all mice were euthanized by intraperitoneal injection of an overdose of pentobarbital (200 mg/kg). Death was confirmed by the absence of a heartbeat and fixed pupils. The tibia samples of all mice were collected, and the surrounding soft tissue was cleaned for subsequent experimental tests.

## Axial tibial loading model

A self-made tibial axial loading model, which can be scanned using AFM, was used as previously described by *Yang et al. (2018b)*. The mechanical load required on the tibia was generated using piezoelectric actuators (PK2FVF1, Thorlabs, Newton, NJ, USA) (Fig. 2B). A force sensor (L6D21, AVIC Electronic Measuring Instruments Co., Ltd, Han Zhong, China) was used to provide feedback and monitor load amplitude. AFM scanning was carried out each time under 4N axial loading conditions.

## Experimental groups

The adult-control-0N group (*n* = 6) refers to tibia collected from 8-month-old mice after 28 days of normal activity without any intervention, which were subjected to AFM scanning under 0N mechanical loading. The adult-control-4N group (*n* = 6) refers to tibia collected from 8-month-old mice after 28 days of normal activity without any intervention, which were subjected to AFM scanning under 4N mechanical loading. The adult-HLU-0N group refers to tibia harvested from 8-month-old mice following 28 days of HLU treatment, which were subsequently subjected to AFM scanning under 0N mechanical loading. The adult-HLU-4N group refers to tibia harvested from 8-month-old mice following 28 days of HLU treatment, which were subsequently subjected to AFM scanning under 4N mechanical loading. The aged-control-0N group (*n* = 6) refers to tibia collected from 17-month-old mice after 28 days of normal activity without any intervention, which were subjected to AFM scanning under 0N mechanical loading. The aged-control-4N group (*n* = 6) refers to tibia collected from 17-month-old mice after 28 days of normal activity without any intervention, which were subjected to AFM scanning under 4N
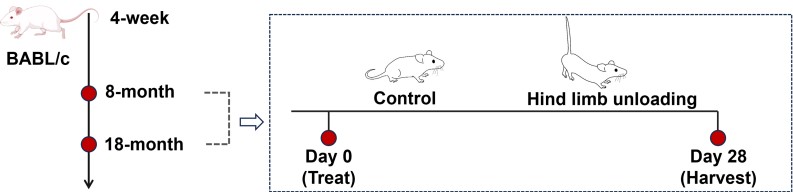

**Figure 1 Schematics representing the experimental design.** Image Source Credit: Figdraw (ID: SOUUI51a8a).

mechanical loading. The aged-HLU-0N group refers to tibia harvested from 17-month-old mice following 28 days of HLU treatment, which were subsequently subjected to AFM scanning under 0N mechanical loading. The aged-HLU-4N group refers to tibia harvested from 17-month-old mice following 28 days of HLU treatment, which were subsequently subjected to AFM scanning under 4N mechanical loading.

The current experimental groups were compared with previously published data from *Liu et al. (2022)*, which examined adult and aged mice under baseline (0 N) loading conditions using identical experimental protocols. Although the studies employed the same treatment procedures and outcome measurements, they were conducted by different research personnel. To ensure a robust comparison, data from four identically processed groups (adult-control-0N, adult-HLU-0N, aged-control-0N, and aged-HLU-0N) were incorporated from a prior study, with all experimental parameters maintained consistently across studies (*Liu et al., 2022*). These datasets provide established references for collagen network responses to both aging and mechanical disuse in bone, enabling direct comparison with new experimental results from additional mouse cohorts.

## AFM image scanning and analysis

The anterior medial tibial surface of mice was selected for AFM scanning because of its flat morphology. Sample preparation involved sequential polishing with 3,000-, 5,000-, and 7,000-grit sandpaper, followed by a 0.5 μm diamond suspension. After ultrasonic cleaning in ultrapure water (5 min), the samples were demineralized with 0.5 M ethylenediaminetetraacetic acid (EDTA, pH 8.0, 30 °C, 15 min) with constant agitation at 200 rpm on an orbital shaker and rinsed with ultrapure water (5 min). This demineralization cycle was repeated thrice (Fig. 2A). Critically, this protocol preserves the intrinsic mechanical properties of tibial bone, as confirmed by prior studies (*Balooch et al., 2008*).

Bone specimens (*n* = 6 per group) were scanned using a PicoPlus 5500 AFM (Agilent, Santa Clara, CA, USA) under ambient conditions (temperature: 22 ± 1 °C; relative humidity: 40 ± 5%). Tapping-mode imaging was performed using a silicon probe (PPP-NCL-20, Nanosensors, USA; nominal resonant frequency: 146–236 kHz; nominal spring constant: 21–98 N/m; actual spring constants were calibrated *via* the thermal tune method). The scan settings were as follows: scan size: 5 μm × 5 μm; scan rate: 1 Hz; pixel resolution: 1,024 × 1,024 pixels; free-air amplitude: ~1.5 V; set point amplitude ratio: 80%

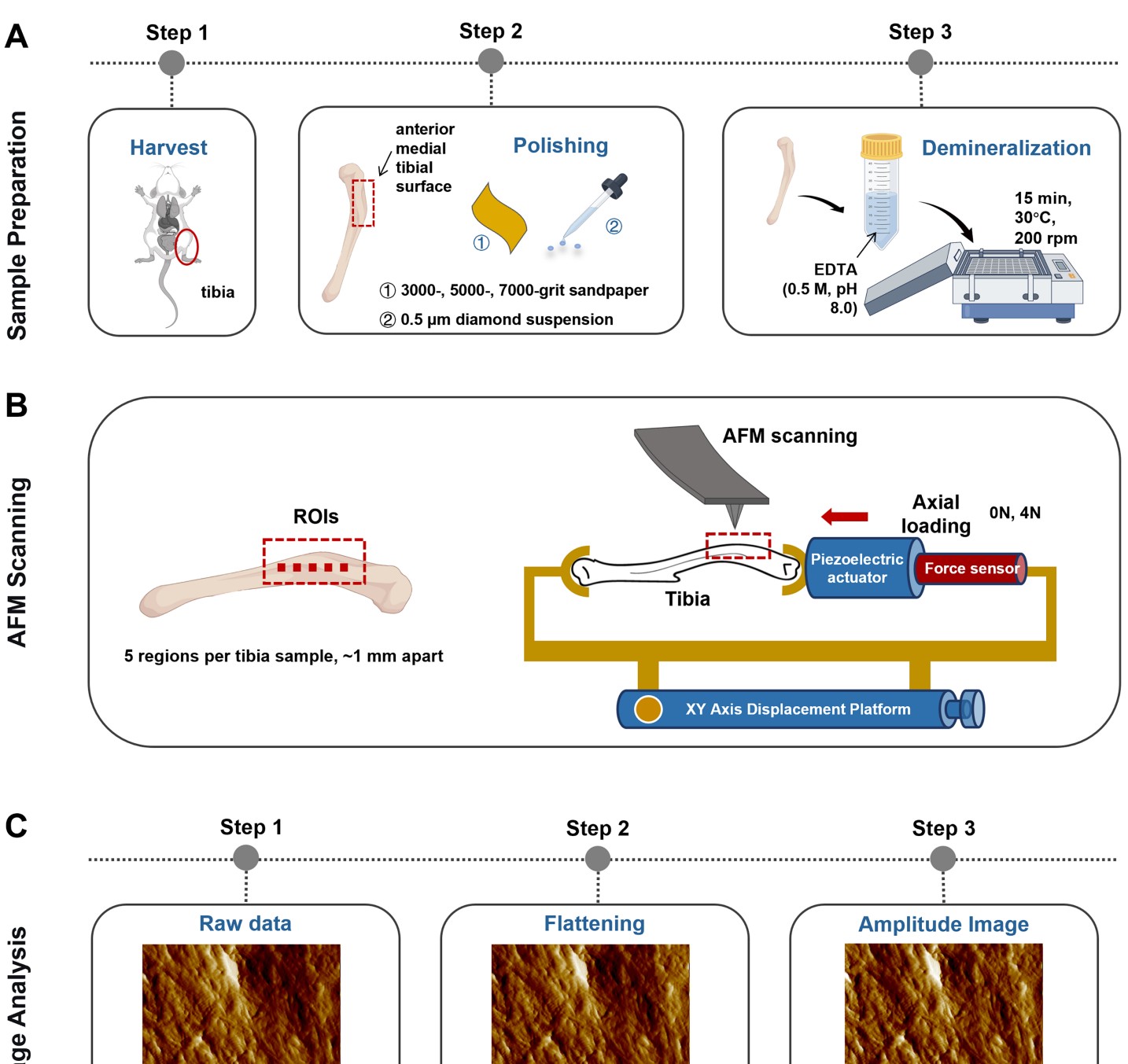

**Figure 2** Schematics representing the sample preparation (A), the axial loading model employed for AFM scanning *in situ* testing (B) and the AFM Image Analysis protocol (C). Image Source Credit: Figdraw (ID: SOUUI51a8a).

of free-air amplitude; integral and proportional gains were optimized in real-time to maintain stable feedback.

The selection of regions of interest (ROI) was performed using a standardized, semi-automatic protocol to ensure consistency and minimize operator bias. For each tibia, the ROIs were predefined along the long axis of the anterior medial surface, spaced approximately 1 mm apart ($n = 5$ per mice). This specific anatomical location was chosen for its relative flatness and reproducibility. Within each predefined ROI, the exact scanning position was manually fine-tuned by the operator to locate regions with clearly exposed and intact collagen fibrils, avoiding areas with large scratches, debris, or artifacts (Fig. 2B). All raw AFM height and amplitude images underwent a standardized post-processing sequence (Fig. 2C). First, a second-order flattening algorithm was applied to each scan line using PicoView software (Agilent, Santa Clara, CA, USA) to remove background tilt and bow while preserving topographical features. Subsequently, a mild low-pass filtering (Gaussian kernel, $\sigma = 0.8$ pixels) was selectively applied exclusively to amplitude images for enhanced visual clarity in figures, but never to height images or any data used for quantitative analysis.

### In situ elastic modulus of the MCFs

After obtaining the morphological images of MCFs, the AFM was switched to contact mode. In situ radial mechanical properties of the exposed MCFs were measured using the AFM-based nanoindentation method under each loading condition. The same scanning probe (PPP-NCL-20, Nanosensors, Neuchatel, Switzerland) was used with an inverted pyramidal tip, which had a side angle of 20°. Using the same image scanning procedure, 20 sites of each image were selected for the nanoindentation test. Before each test, the range deviation signal was calibrated by pushing the probe onto the mica plate. To reduce the viscosity effect, indentation was performed at a low speed of piezoelectric displacement (2.5 μm/s) for 4.8 s (24,000 sampling points). The force-displacement curve was recorded during the indentation.

### AFM image and force-displacement curve analyses
#### D-periodic spacing analysis of MCFs

The D-periodic spacing of MCFs was quantified based on AFM amplitude images. For each scanning site, ten individual MCFs were manually traced along their longitudinal axes using PicoView software (Agilent, Santa Clara, CA, USA). Fibril periodicity was determined by measuring peak-to-peak distances in amplitude deviation curves using a custom MATLAB algorithm (The MathWorks, Natick, MA, USA).

### MCF orientation analysis

Fibril orientation was assessed from each AFM image by randomly selecting twenty mineralized collagen fibrils (MCFs) using a combined approach of automated segmentation and random sampling. More specifically, AFM amplitude images were processed in Fiji/ImageJ (NIH, Bethesda, MD, USA) using the "Trainable Weka Segmentation" plugin, which was trained to identify MCFs based on manual input. The

labeled results were exported, and automatically identified MCFs by assigning random numbers. A custom-written MATLAB routine (The MathWorks, Inc., Natick, MA, USA) was then used to randomly select twenty MCFs from each image. The angle between the longitudinal axis of each fibril and the horizontal reference line (corresponding to the AFM image axis) was measured. To account for variations in sample positioning, all angular values were normalized to the mean orientation. The resulting relative orientation angles represent the deviation from the predominant fibril alignment direction and were used to establish the relationship between the tibial long axis and the AFM scanning axis.

### Radial elastic modulus of the MCFs

The classical Hertz model was used to analyze the elastic modulus of the site. For the conical probe, the model satisfied the following equation:

$$F = \frac{2}{\pi} \frac{E}{1-v^2} \tan \alpha \delta^2$$

where $F$ is the indentation force, $E$ is the elastic modulus of the collagen fibril, $V$ is Poisson's ratio (set as 0.3), $\alpha$ is the half-open angle of the probe (in this test, the half-open angle of the probe was 10°), and $\delta$ is the indentation distance. The force-displacement curve was converted into a force-indentation curve. The Hertzian contact model (*Lin, Dimitriadis & Horkay, 2007*; *Carl & Schillers, 2008*) was adopted to fit the selected force-indentation curve with another custom-written MATLAB (The MathWorks Inc., Natick, MA, USA) routine (*Yang et al., 2018a*). The elastic modulus of the pressure part was then calculated and analyzed statistically.

### Statistics

Results are presented as the mean ± standard deviation. All statistical analyses were performed using GraphPad Prism 8 (GraphPad Software, San Diego, CA, USA). The D-periodic spacing distribution of collagen fibrils was characterized using histogram plots and cumulative distribution function analysis (*Wallace et al., 2011*). Fibril orientation angles were normalized and plotted as frequency distribution functions. For comparative analysis, Holm-Sidak's multiple comparisons test was used to evaluate age- and load-dependent variations in bone structural/mechanical properties, and Kolmogorov-Smirnov (K-S) test was used to assess differences in D-periodic spacing and orientation angle distributions before loading *vs.* after loading. A threshold of $p \le 0.05$ was considered statistically significant.

## RESULTS

### Morphology of MCFs in both adult and aged bone

AFM amplitude imaging revealed clear morphological features of MCFs across the experimental groups, allowing for qualitative comparison of the fibrils organization (Figs. 3A–3H). The images that exposed the clearly visible collagen fibrils were used for subsequent quantitative analysis of D-periodic spacing and fibril orientation in all groups.

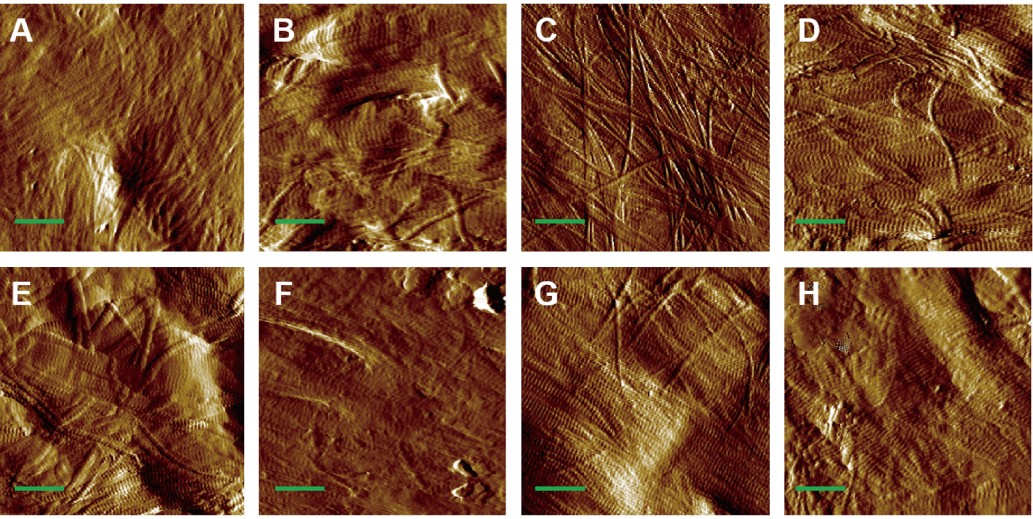

**Figure 3 AFM amplitude image of the MCFs in murine cortical tibia ($n = 6$).** (A) The adult-control-0N group. (B) the adult-HLU-0N group. (C) the adult-control-4 N group. (D) the adult-HLU-4N group. (E) the aged-control-0N group. (F) the aged-HLU-0N group. (G) the aged-control-4 N group. (H) the aged-HLU-4N group. Data for the 0N groups (A, B, E and F) were derived from our previous publication (*Liu et al., 2022*). Green bar: 1 μm.                 

## Age- and mechanical loading-dependent D-periodic spacing dynamics

D-periodic spacing analysis revealed significant age-related differences (Fig. 4). For the adult group, the D-periodic spacing of MCFs remained unchanged under different mechanical loading conditions (70 ± 9 nm at 0 N *vs.* 70 ± 9 nm at 4 N, $p = 0.87$). In contrast, for the aged group, the D-periodic spacing of MCFs revealed 10% and 8.6% larger D-spacing than that of the adult group, respectively (77 ± 11 nm at 0 N *vs.* 76 ± 12 nm at 4 N, $p < 0.001$). Notably, disuse induced different effects: the adult-HLU at 4 N showed 7% D-spacing increase than that at 0 N (71 ± 13 nm at 0 N *vs.* 76 ± 7 nm at 4 N, $p < 0.001$), whereas aged-HLU exhibited equal mean D-periodic spacing under different loading conditions (74 ± 11 nm at 0 N *vs.* control 77 ± 11 nm at 4 N, $p = 0.23$).

The overall mean D-periodic spacing in aged groups was significantly larger than that in adult groups ($p < 0.001$). The adult-control group showed no significant D-periodic spacing variations between loading conditions, whereas a contrasting response was observed in aged specimens. Mechanical loading with 4 N induced slight but significant D-periodic spacing reduction in aged controls ($p < 0.01$). Disuse induced divergent aging-dependent responses–adult groups exhibited significant D-periodic spacing elevation post-disuse ($p < 0.001$); conversely, aged groups demonstrated paradoxical spacing contraction ($p < 0.01$). Notably, mechanical loading failed to exert significant effects in the spacing parameters of either age group.

MCFs distribution analysis demonstrated fundamental age-related differences. While 55% of MCFs in the adult controls exceeded 70 nm D-periodic spacing, their aged counterparts showed only 28% in this range (Figs. 4C, 4D). Statistical modeling identified

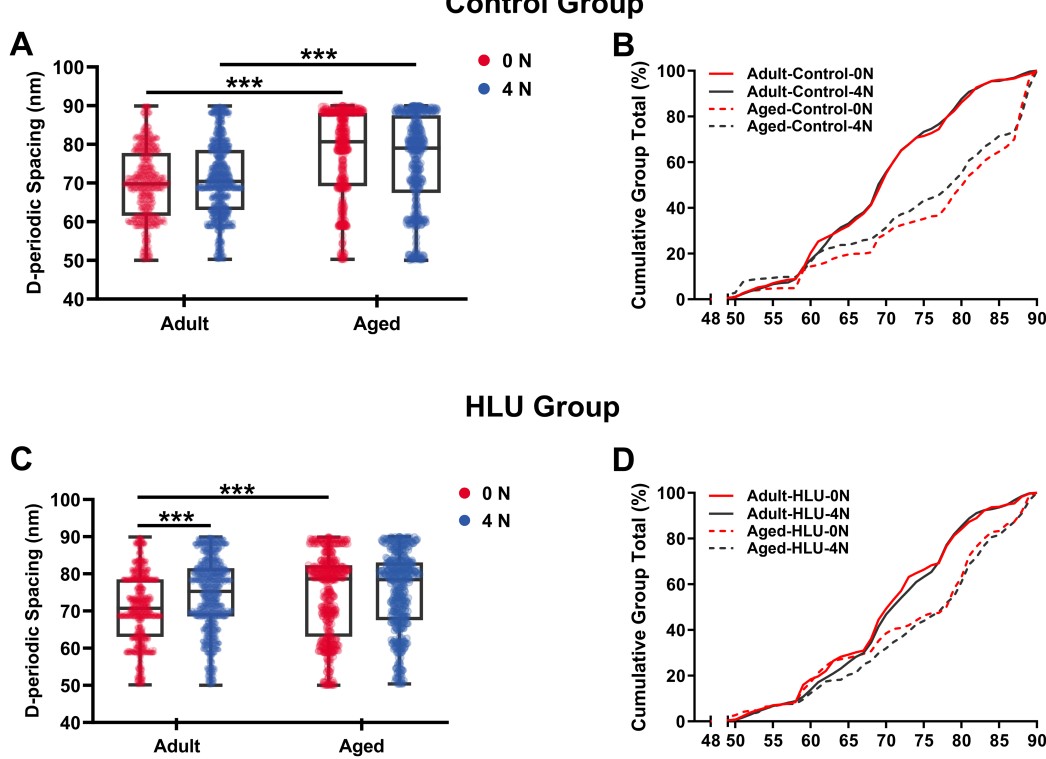

**Figure 4 D-periodic spacing and population distribution of the MCFs ($n$ = 6 per group).** Boxplot graph of mean values of D-periodic spacing of the MCFs for the control (A) and HLU (C) group. The cumulative density function of D-periodic spacing of the MCFs of the control (B) and HLU (D) group. ***: $p < 0.001$.

significant main effects of age ($p < 0.001$) and disuse ($p < 0.01$), with marked age-disuse interaction ($p < 0.001$) with both spacing parameters and MCF distribution patterns.

Cumulative distribution profiling (Figs. 4C, 4D) revealed impaired mineralization homogeneity in aged groups as evidenced by widened Kolmogorov-Smirnov distances (adult-control: 0.12 *vs*. aged-control: 0.28). This quantifies enhanced structural heterogeneity in aged bone matrices, suggesting age-dependent dysregulation of mineral incorporation processes.

## Orientation of MCFs

The orientation distribution of the MCFs between different age groups, or different loading condition groups for the control did not differ (Figs. 5A, 6A, 6B, 6C). The adult-HLU-0 N and aged-HLU-4 N showed a significant difference in the MCF orientation range of 0–10° ($p < 0.05$) and 30–40° ($p < 0.05$). Therefore, disuse and aging significantly contribute to the disorientation of MCFs (Figs. 5B, 6D).

## *In situ* radial elastic modulus of collagen fibrils

With mechanical loadings of 0 and 4 N, the radial elastic moduli of MCFs in the adult-control group were 0.30 ± 0.13 and 0.39 ± 0.13 GPa, while those in the aged-control

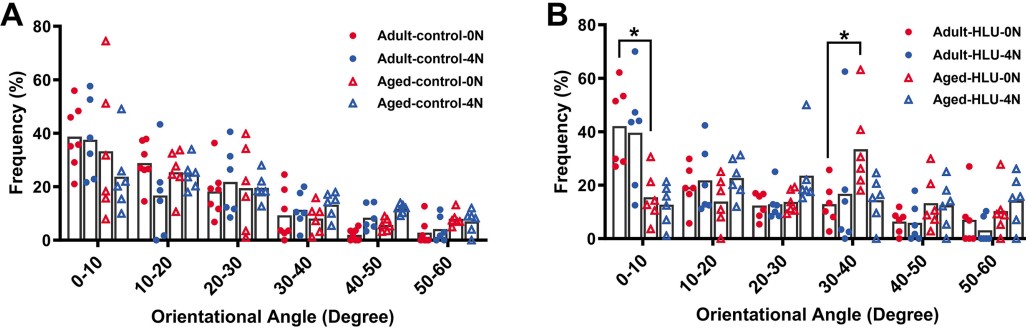

**Figure 5 Distribution frequency of normalized orientation angle of MCFs in cortical bone of mice under different loading conditions (*n* = 6 per group).** Kolmogorov-Smirnov tests failed to indicate significant differences between two loading conditions for both the control and disuse groups. (A) the control group. (B) the HLU group. *: *p* < 0.05.

group were 0.74 ± 0.17 and 0.67 ± 0.15 GPa, respectively. After disuse, the radial elastic modulus of the adult-HLU group was 0.32 ± 0.11 and 0.33 ± 0.09 GPa when the MCFs were under 0 and 4 N mechanical loading, and that in the aged-HLU group was 0.77 ± 0.23 and 0.75 ± 0.29 GPa, respectively. The radial elastic modulus of the aged group was significantly higher than that of the adult group. With increased mechanical loading, the elastic modulus of the adult control group increased (*p* < 0.001), and the elastic modulus of the aged control group decreased (*p* < 0.001). The main effects of age and mechanical environment on radial elastic modulus were determined (*p* < 0.001 for age, *p* < 0.001 for mechanical environment, *p* < 0.001 for the interaction of age and mechanical environment).

## DISCUSSION

In the present study, experimental animal models of aging and disuse were established, combined with an axial loading bone scanning system. MCFs in the cortical tibiae of adult and aged mice were systematically characterized at the nanoscale under varying mechanical loading conditions. Specifically, the D-periodic spacing, orientation, and *in situ* radial elastic modulus of MCFs were quantified across different loading amplitudes, and the differential bone deformation mechanisms between adult and aged groups were elucidated. With increasing mechanical loads, the D-periodic spacing of MCFs remained unchanged in both adult and aged cohorts. However, early reorientation of MCFs was observed in adult mice during bone deformation, whereas the aged group exhibited greater structural stability. Notably, the radial elastic modulus of adult bone increased progressively during deformation, whereas a gradual decline was detected in aged bone. Disuse exacerbated mechanical sensitivity, particularly in aged mice. These findings demonstrate that age significantly influences the nanomechanical behavior of MCFs under load. Distinct deformation mechanisms were identified between age groups, which are primarily governed by MCF stretching and sliding during elastic deformation. Application of mechanical loading models is a widely adopted approach for investigating load-induced bone adaptation. Among these, the axial tibial loading model represents one of the most

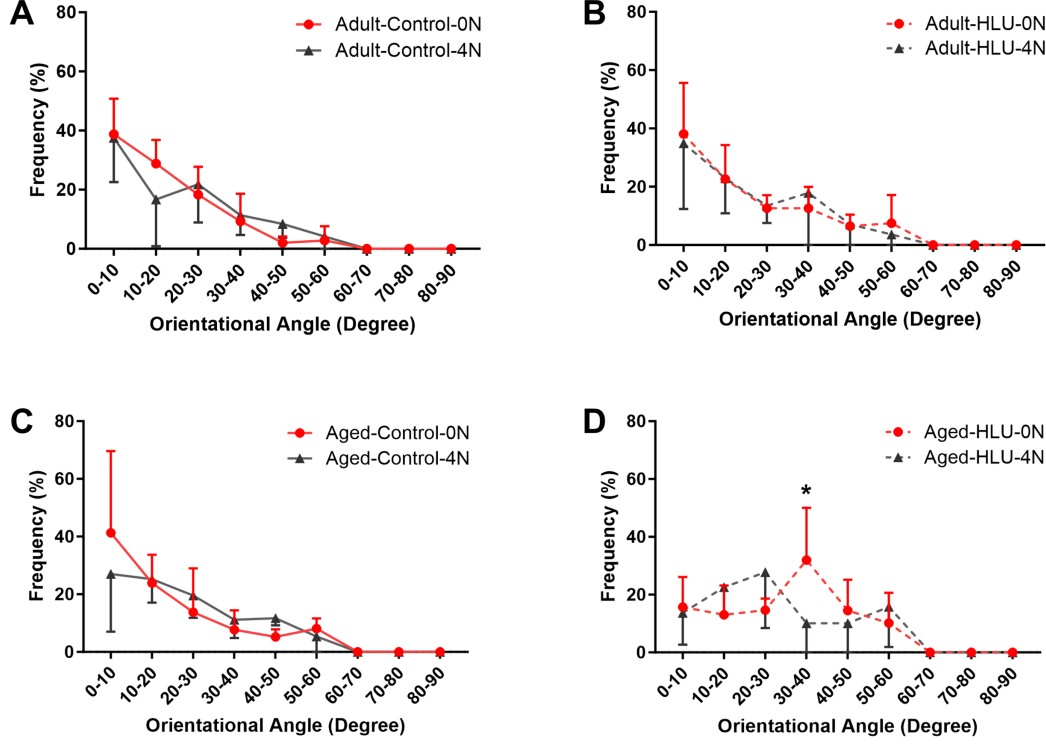

**Figure 6** The line graph of the directional distribution of collagen fibrils between 0 group and 4 N group calculated from the AFM amplitude image for the (A) adult control, (B) adult HLU, (C) aged control and (D) aged HLU. *$p < 0.05$.

frequently utilized methods for simulating near-physiological mechanical conditions, as demonstrated in previous studies (*Yang et al., 2018b*). In this work, a custom mechanical loading device, compatible with AFM, was used to apply precise and quantifiable axial loads to the tibia while simultaneously enabling real-time scanning of its ultrastructural surface morphology. Finite element simulations confirmed that tibial deformation remained within the elastic regime at loads below 6 N (*Yang et al., 2018a*). Consequently, under the 2 and 4 N loading conditions applied in this study, MCFs in the tibia underwent purely elastic deformation. As the fundamental structural units enabling the physiological function of collagen, MCFs self-assemble into a three-dimensional network that confers both mechanical strength and elasticity to bone tissue, critically influencing its ductility. Previous studies have demonstrated that MCFs sustain elongation exceeding 30% while exhibiting ultimate tensile strength within the 10–100 MPa range.

While disuse undoubtedly exacerbates bone fragility, it is critical to emphasize that aging itself exerts profound and covariant effects on the nanoscale properties of bone, which underpin the observed mechanical degradation. The present findings indicate that aging, independent of disuse, is associated with significant alterations in MCF organization and mechanics. Specifically, the aged bone exhibited increased D-periodic spacing heterogeneity and a broader distribution of fibril orientations even under control conditions (Figs. 4B, 4D), suggesting an age-related loss of structural homogeneity. This

inherent age-induced disorganization likely stems from the non-enzymatic accumulation of AGEs within the collagen matrix, which alters fibrillar cross-linking and enhances brittleness. Furthermore, the observed shift in the deformation mechanism, from fibril stretching in adults to immediate sliding in aged bone, directly reflects an aging-related deterioration in the quality of the organic matrix and its interface with the mineral phase. Consequently, the aged bone matrix presents a mechanically compromised baseline state. Disuse, when superimposed on this pre-existing age-related vulnerability, acts as a second hit, further amplifying the nanoscale mechanical deficits and leading to the pronounced sensitivity we observed in the aged-HLU group. Therefore, the interplay between aging and disuse is not merely additive but synergistic, wherein aging establishes a susceptible nanoscale environment that dramatically potentiates the detrimental effects of mechanical unloading.

The D-periodic spacing of MCFs, typically reported as ~67 nm, has been widely utilized as a morphological indicator (*Sizeland et al., 2020*; *Wallace, 2015*). Under tensile loading, relative sliding occurs between the collagen and mineral phases, resulting in D-spacing values significantly exceeding the nominal 67 nm. In this study, periodic MCF structures in adult and aged murine bone were clearly resolved using AFM (Fig. 3). Previous studies have demonstrated that MCF elongation accompanies mineral deposition, leading to increased D-periodic spacing (*Wallace, 2015*). As our experimental samples were obtained from adult and aged mice with particularly high mineralization levels in the aged cohort, enhanced mineral deposition with aging was reflected as greater D-periodic spacing. Quantitative analysis revealed an average D-periodic spacing of 70 ± 9 nm in adult bone, slightly exceeding previously reported values, while the aged group exhibited even larger spacing (77 ± 11 nm) (Figs. 4A, 4C), consistent with their elevated mineralization state. Distribution analysis indicated that most adult fibrils fell within the 65–75 nm range (Fig. 4B), whereas the aged group displayed a broader distribution (60–85 nm) (Fig. 4D). Cumulative density functions further highlighted differences between the loaded and unloaded groups. Notably, the supramolecular fibrillar architecture remained intact throughout the stretch-release cycles. This stability primarily arises from the integrity of enzymatic cross-links within and between collagen molecules, which is fundamental for preserving fibrillar structure under strain. The mineral phase in bone likely provides additional reinforcement to the collagen fibrils network. While single fibrils from rat tail tendon can be stretched without breaking (*Gachon & Mesquida, 2021*), mineralization provides supplemental reinforcement that enhances resistance to structural damage.

Simultaneously, variations in MCF orientation were characterized based on the scanned images (Figs. 5, 6). MCFs, embedded within extrafibrillar mineral particles, were observed to assemble into fibril arrays (*Currey, 2005*). In lamellar bone, these arrays formed parallel fibril sheets organized in a rotating plywood-like pattern (*Weiner et al., 1997*; *Schwiedrzik et al., 2014*). Each sub-layer exhibited distinct fibril twist angles relative to the long axis of the bone, resulting in imperfect alignment with the tibial longitudinal axis. Consequently, under uniaxial tensile loading, heterogeneous fibril responses were observed, explaining the MCF dislocation phenomena detected in this study. MCF orientation critically influenced local bone matrix adaptation to mechanical stimuli. Quantitative analysis of orientation distribution, performed using an objective visual detection method with

mature models to eliminate human bias, revealed load-dependent responses. In the adult bone, the MCF alterations involved a sequence of fibril stretching followed by fibril sliding relative to each other. By contrast, the aged bone exhibited MCF sliding at the onset of mechanical loading, but without a distinct stretching phase. At low loads, intermolecular deformation remained within elastic limits, preserving MCF mechanical integrity. Longitudinally oriented MCFs demonstrated superior stiffness and deformation resistance (*Stockhausen et al., 2021*). Our data indicate a predominant small-angle orientation distribution (0–10°; Fig. 5B), aligning with these mechanical advantages. Notably, aged MCFs showed enhanced orientational stability under load, potentially mediated by mineral-facilitated interfibrillar sliding. Demineralization experiments suggested that extrafibrillar minerals regulate MCF organization as their removal exacerbated load-induced disorganization. Under identical demineralization conditions, MCFs showed variable stratification (single *vs.* multilayer), further explaining their orientation heterogeneity.

Bone represents a hierarchically organized composite material composed of a mineralized collagen matrix, whose mechanical properties are determined by the interplay between inorganic minerals and organic proteins. The organic matrix primarily contributes to bone hardness. Previous studies have reported that type I collagen fibers from the bovine Achilles tendon exhibited Young's moduli of 1.9 ± 0.5 GPa (dry) and 1.25 ± 0.1 MPa (hydrated) when measured using Hertzian mechanics (*Grant et al., 2008*). Similarly, dried type I collagen fibrils from the rat tail tendon demonstrated Young's moduli ranging from 5–11.5 GPa (*Wenger et al., 2007*). Distinct elastic moduli were identified for overlapping (~2.2 GPa) *vs.* gap regions (~1.2 GPa) in dry type I collagen fibrils from the bovine Achilles tendon (*Kontomaris, Stylianou & Malamou, 2022*; *Heim, Matthews & Koob, 2006*). The radial elastic modulus of bone MCFs was found to be strongly dependent on the surrounding mineral environment (*Unal, Creecy & Nyman, 2018*). Extensive evidence has demonstrated that mineralization enhances the elastic behavior of MCFs (*Liu et al., 2014*; *Depalle et al., 2015*; *García-Rodríguez & Martínez-Reina, 2017*), which was corroborated in the present study. Although surface demineralization was performed using EDTA, intrafibrillar minerals remained preserved, maintaining MCF structural integrity. Notably, aged bone tissue exhibited an approximately 2.5-fold higher MCF elastic modulus compared to that of the adult tissue (Fig. 5B), consistent with its elevated mineralization state. Raman spectroscopy confirmed significantly higher mineral-to-matrix ratios in aged specimens in our previous study (*Liu et al., 2022*), indicating progressive mineral accumulation during aging. Increasing mechanical loads to 4 N led to progressive loosening of mineral-collagen interactions, resulting in a decreased elastic modulus in the aged bone. By contrast, in the adult bone, the MCF elastic modulus showed load-dependent increase (Fig. 7A), which were likely attributable to tensile stretching of MCFs.

The study systematically characterized the response of MCFs during bone loss, elucidating their role in deformation mechanisms under abnormal mechanical conditions. Bone disuse significantly reduced bone mineral density and altered MCF mineralization states. Despite the intrinsic linkage of D-periodic spacing with collagen molecular

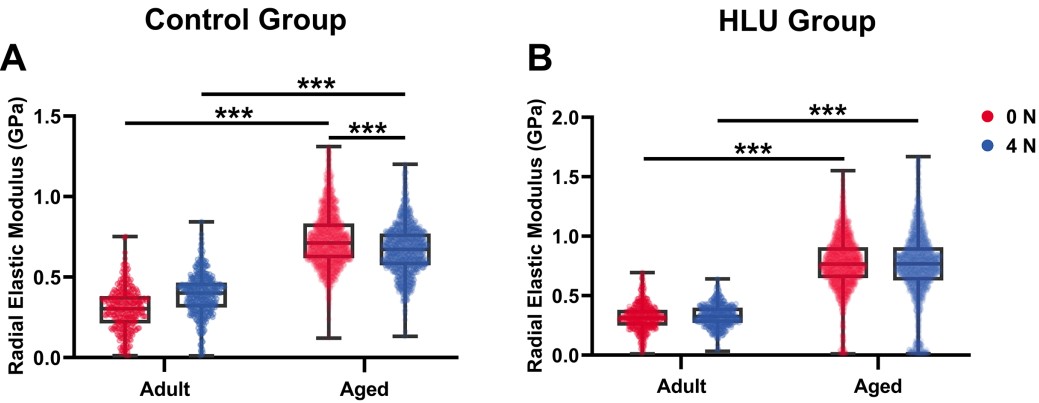

**Figure 7** Histogram of *in situ* radial elastic modulus of the MCFs in cortical tibia in the control group (A) and the HLU group (B) under different mechanical loading conditions (*n* = 6 per group). ***: *p* < 0.001.

assembly (*Garnero, 2015*), no significant alterations in MCF D-periodicity were observed, suggesting preserved inter-molecular hydrogen bonding. These findings align with previous reports showing stable D-spacing in osteogenesis imperfecta models (*Fang & Holl, 2013*). However, mechanical loading induced significant D-periodic variations in both adult and aged mice compared to those in the healthy controls (Fig. 4). Hindlimb unloading-mediated disuse modified both intra- and extra-fibrillar mineral distributions (*Yang et al., 2019*), altering the MCF prestress states and redistributing load-bearing capacity between collagen and mineral phases (*Nair, Gautieri & Buehler, 2014*). Disuse increased D-periodic spacing in adult MCFs without affecting orientation, whereas in aged specimens, susceptibility was increased with reduced D-spacing (74 ± 11 nm) likely attributable to advanced bone loss. This structural alteration rendered the aged MCFs more responsive to mechanical loading. Notably, the disused aged bone demonstrated enhanced mechanical sensitivity, as loading induced greater disorganization in MCF orientation (Fig. 6D). This increased orientational dispersion correlated with elevated elastic modulus values (Fig. 7B). The mechanical consequences of such structural changes were interpreted through the lens of the hierarchical organization of bone, where ordered (aligned fibrils) and disordered (entangled) phases differentially influence mineralization and bulk mechanics (*Holmes et al., 2018*; *Tertuliano & Greer, 2016*). In aged bone with slowed metabolism, modulus enhancement was primarily attributed to intensified MCF mineralization. However, 4N loading caused mineral detachment from MCFs, reducing both modulus and D-spacing. During deformation, MCF crosslinking was found to resist loading through both intrinsic stiffness and intermolecular connectivity (*Balooch et al., 2008*; *Pang et al., 2020*), but no significant correlation was found between crosslinking density and bone loss severity.

In this study, AFM was used to characterize the morphological properties of MCFs on the cortical bone surface of murine proximal tibiae. Before AFM scanning, EDTA-mediated demineralization was performed without affecting the mechanical response of MCFs during bone deformation. To ensure consistency, demineralization

protocols were carefully standardized within each age cohort to account for inherent variations in bone mineralization levels. Although previous work demonstrated that demineralization altered the mechanical properties of human dental MCFs (*Balooch et al., 2008*), the current investigation focused specifically on tissue-level deformation responses rather than intrinsic MCF modifications, thereby minimizing the potential confounding effects from demineralization. Several methodological considerations were addressed: (1) potential measurement artifacts arising from instrumental factors, scanning environments, and analytical software were corrected through standardized calibration protocols using reference samples; (2) although AFM sample preparation was relatively straightforward, dehydration-induced fibril shrinkage was considered negligible under ambient conditions, as significant dimensional changes only occurred during complete vacuum dehydration; and (3) D-periodic spacing remained stable during air-drying procedures.

However, a few limitations of the present study must be mentioned. Firstly, the resolution of the AFM images, while adequate for qualitative assessment and orientation analysis, was not ideal for precise quantification of nanoscale features of MCF, such as D-periodic spacing. This limitation likely contributed to the observed variability in D-periodic spacing measurements and underscores the need for higher-resolution imaging in future work. Secondly, the decision to demineralize the bone surface to expose collagen fibrils precluded the analysis of the native mineral phase. Consequently, the direct effects of aging and disuse on mineral morphology and its contribution to the altered mechanical behavior could not be assessed. Thirdly, the application of AFM in the present study, while powerful for nanoscale surface assessment of intact bone tissue, is inherently limited to two-dimensional analysis. Complementary volumetric imaging techniques, such as electron microscopy or synchrotron X-ray scattering, would be beneficial to be incorporated in future work to provide multiscale structural insights.

## CONCLUSIONS

In summary, this study revealed that MCFs, as fundamental structural components of the bone matrix, exhibit distinct age-dependent mechanical adaptation mechanisms during elastic deformation. Although the D-periodic spacing of MCFs remained stable under mechanical loading in both age groups, adult bone demonstrated initial fibril stretching characterized by collagen reorientation and increased radial elastic modulus, followed by intermolecular sliding at higher loads. In contrast, aged bone maintained greater structural stability through predominant intermolecular sliding, accompanied by progressive modulus reduction. Notably, disuse exacerbated these mechanical responses, particularly in aged bone, wherein deformation initiated through fibril stretching before transitioning to molecular sliding. Overall, these findings provide critical nanoscale evidence for how MCF-level adaptations govern the tissue-level mechanical behavior of bone, elucidating fundamental mechanisms through which aging and disuse impair bone mechanical integrity. By bridging the gap between fibrillar reorganization and macroscopic bone properties, this work advances our understanding of age-related bone deterioration and offers potential insights for developing targeted therapeutic strategies.

## ACKNOWLEDGEMENTS

Thanks to Dr. Qiufeng Wang, Dr. Dan Feng and Dr. Yixue Li from the Analytical & Testing Center of Northwestern Polytechnical University, and Dr. Jia Liu at the Instrument Analysis Center of the Xi'an Jiaotong Univeristy for their invaluable assistance during the experiments.

### Funding

This research was funded by the projects of the National Natural Science Foundation of China (Grant numbers 12272317 and 12302403), the Space Medical Experiment Project of CMSP (HYZHXMH01006), and the Foreign Expert Project (Grant number QN2023183004L). The funders had no role in study design, data collection and analysis, decision to publish, or preparation of the manuscript.

### Grant Disclosures

The following grant information was disclosed by the authors:
National Natural Science Foundation of China: 12272317, 12302403.
Space Medical Experiment Project of CMSP: HYZHXMH01006.
Foreign Expert Project: QN2023183004L.

### Competing Interests

The authors declare that they have no competing interests.

### Author Contributions

- Fan Li conceived and designed the experiments, performed the experiments, analyzed the data, authored or reviewed drafts of the article, and approved the final draft.
- Fa Liu conceived and designed the experiments, performed the experiments, analyzed the data, prepared figures and/or tables, authored or reviewed drafts of the article, and approved the final draft.
- Chenxi Ren performed the experiments, authored or reviewed drafts of the article, and approved the final draft.
- Shuyang Zhang analyzed the data, prepared figures and/or tables, authored or reviewed drafts of the article, and approved the final draft.
- Zhe Wang analyzed the data, authored or reviewed drafts of the article, and approved the final draft.
- Pengfei Yang conceived and designed the experiments, authored or reviewed drafts of the article, and approved the final draft.

### Animal Ethics

The following information was supplied relating to ethical approvals (*i.e.*, approving body and any reference numbers):

Animal Ethics and Welfare Committee at Northwestern Polytechnical University (No. 2018036).

## Data Availability

The raw data are available in the Supplemental Files.

## Supplemental Information

Supplemental information for this article can be found online at http://dx.doi.org/10.7717/peerj.20313#supplemental-information.

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

# PeerJ

**Sasaki N, Odajima S. 1996.** Stress-strain curve and Young's modulus of a collagen molecule as determined by the X-ray diffraction technique. *Journal of Biomechanics* **29(5)**:655–658 DOI 10.1016/0021-9290(95)00110-7.

**Schwiedrzik J, Raghavan R, Bürki A, LeNader V, Wolfram U, Michler J, Zysset P. 2014.** In situ micropillar compression reveals superior strength and ductility but an absence of damage in lamellar bone. *Nature Materials* **13(7)**:740–747 DOI 10.1038/nmat3959.

**Shen ZL, Dodge MR, Kahn H, Ballarini R, Eppell SJ. 2008.** Stress-strain experiments on individual collagen fibrils. *Biophysical Journal* **95(8)**:3956–3963 DOI 10.1529/biophysj.107.124602.

**Silver FH, Jaffe M, Shah RG. 2018.** 11-Structure and behavior of collagen fibers. In: Bunsell AR, ed. *Handbook of Properties of Textile and Technical Fibres.* Second Edition. New Delhi: Woodhead Publishing, 345–365.

**Sizeland KH, Wells HC, Kirby NM, Hawley A, Mudie ST, Ryan TM, Haverkamp RG. 2020.** Bovine meniscus middle zone tissue: measurement of collagen fibril behavior during compression. *International Journal of Nanomedicine* **15**:5289–5298 DOI 10.2147/ijn.S261298.

**Stockhausen KE, Qwamizadeh M, Wölfel EM, Hemmatian H, Fiedler IAK, Flenner S, Longo E, Amling M, Greving I, Ritchie RO, Schmidt FN, Busse B. 2021.** Collagen fiber orientation is coupled with specific nano-compositional patterns in dark and bright osteons modulating their biomechanical properties. *ACS Nano* **15(1)**:455–467 DOI 10.1021/acsnano.0c04786.

**Stylianou A, Kontomaris S-V, Grant C, Alexandratou E. 2019.** Atomic force microscopy on biological materials related to pathological conditions. *Scanning* **2019(12)** DOI 10.1155/2019/8452851.

**Svensson RB, Hassenkam T, Hansen P, Magnusson SP. 2010.** Viscoelastic behavior of discrete human collagen fibrils. *Journal of the Mechanical Behavior of Biomedical Materials* **3(1)**:112–115 DOI 10.1016/j.jmbbm.2009.01.005.

**Tertuliano OA, Greer JR. 2016.** The nanocomposite nature of bone drives its strength and damage resistance. *Nature Materials* **15(11)**:1195 DOI 10.1038/nmat4719.

**Unal M, Creecy A, Nyman JS. 2018.** The role of matrix composition in the mechanical behavior of bone. *Current Osteoporosis Reports* **16(3)**:205–215 DOI 10.1007/s11914-018-0433-0.

**Ushiki T. 2002.** Collagen fibers, reticular fibers and elastic fibers. A comprehensive understanding from a morphological viewpoint. *Archives of Histology and Cytology* **65(2)**:109–126 DOI 10.1679/aohc.65.109.

**van der Rijt JAJ, van der Werf KO, Bennink ML, Dijkstra PJ, Feijen J. 2006.** Micromechanical testing of individual collagen fibrils. *Macromolecular Bioscience* **6(9)**:697–702 DOI 10.1002/mabi.200600063.

**Wallace JM. 2015.** Effects of fixation and demineralization on bone collagen D-spacing as analyzed by atomic force microscopy. *Connective Tissue Research* **56(2)**:68–75 DOI 10.3109/03008207.2015.1005209.

**Wallace JM, Orr BG, Marini JC, Holl MMB. 2011.** Nanoscale morphology of Type I collagen is altered in the Brtl mouse model of Osteogenesis Imperfecta. *Journal of Structural Biology* **173(1)**:146–152 DOI 10.1016/j.jsb.2010.08.003.

**Weiner S, Arad T, Sabanay I, Traub W. 1997.** Rotated plywood structure of primary lamellar bone in the rat: orientations of the collagen fibril arrays. *Bone* **20(6)**:509–514 DOI 10.1016/s8756-3282(97)00053-7.

**Wenger MPE, Bozec L, Horton MA, Mesquida P. 2007.** Mechanical properties of collagen fibrils. *Biophysical Journal* **93(4)**:1255–1263 DOI 10.1529/biophysj.106.103192.

**Yang PF, Huang LW, Nie XT, Yang Y, Wang Z, Ren L, Xu HY, Shang P. 2018b.** Moderate tibia axial loading promotes discordant response of bone composition parameters and mechanical properties in a hindlimb unloading rat model. *Journal of Musculoskeletal & Neuronal Interactions* **18(2)**:152–164.

**Yang PF, Nie XT, Wang Z, Al-Qudsy LHH, Ren L, Xu HY, Rittweger J, Shang P. 2019.** Disuse impairs the mechanical competence of bone by regulating the characterizations of mineralized collagen fibrils in cortical bone. *Frontiers in Physiology* **10**:404 DOI 10.3389/fphys.2019.00775.

**Yang P-F, Nie X-T, Zhao D-D, Wang Z, Ren L, Xu H-Y, Rittvveger J, Shang P. 2018a.** Deformation regimes of collagen fibrils in cortical bone revealed by in situ morphology and elastic modulus observations under mechanical loading. *Journal of the Mechanical Behavior of Biomedical Materials* **79**:115–121 DOI 10.1016/j.jmbbm.2017.12.015.

**Zioupos P, Kirchner HOK, Peterlik H. 2020.** Ageing bone fractures: the case of a ductile to brittle transition that shifts with age. *Bone* **131**:115176 DOI 10.1016/j.bone.2019.115176.