# Peer review of "In situ mechanical behavior of mineralized collagen fibrils in murine cortical bone is altered by aging and disuse"

_PeerJ, doi:10.7717/peerj.20313_

## Round 0.1 · original submission · Major Revisions

· Academic Editor

Major Revisions

Dear Dr. Yang,
Based on two reviews, the editors believe that your manuscript consider requiring a major revision of the article to reach its full potential.

My primary concerns are related to the title, abstract, exclusion of references, and exclusion of detailed explanations of the entire AFM assessment of the bone. Additionally, the study design requires a more detailed explanation to ensure reproducibility and an adequate assessment of data validity. The usage of previously published material should also be stated in the abstract and should be acknowledged in a separate limitations section. I would suggest that all limitations of the study be presented in a separate section of the discussion. The quality of the English language used in the manuscript needs improvement.

**Language Note:** The Academic Editor has identified that the English language must be improved. PeerJ can provide language editing services - please contact us at [email protected] for pricing (be sure to provide your manuscript number and title). Alternatively, you should make your own arrangements to improve the language quality and provide details in your response letter. – PeerJ Staff

Reviewer 1 ·

Basic reporting

The manuscript relies on a previously published data set (see lines 137-145). In addition, all figures present the comparison between 0N and 4N force treatment on the morphology and mechanics of demineralized bone, but in the discussion, the authors keep referring to a 2N force level, even though no data are shown for that treatment.

Experimental design

For the morphological characterization of the bone surface, the authors rely on manual selection of a few fibrils (10-20) and use 5-micron-square images where the D-band is just barely visible. This leads them to report large D-band values (70 nm or more) even at 0N treatment and large standard deviation (9-10 nm). I recommend collecting higher resolution images at the same scan size to improve D-band data. The orientation mapping should also be done using automated software rather than by hand.

Validity of the findings

One finding is not supported by the presented data. Line 210 describes localized buckling of fibrils on one of the images, but I was not able to identify those buckling sites on the provided images.

The statement lines 302 to 304 are incorrect. Mineralization is not necessary for preserving the D-band during stretching. This is the role of the enzymatic crosslinks. See beautiful work by Gachon and Mesquida showing that single fibrils from rat tail tendon can be stretched without breaking while reaching D-band length of close to 80 nm.

In their discussion, the authors keep describing a multi-stage deformation mechanism of the fibrils, but their data only show one force level, so they must be talking about previous studies without making it clear that they are, which is very confusing.

Additional comments

Line 168, the speed should be in micron/s, not m/s.

·

Basic reporting

The manuscript titled "In situ mechanical behavior of mineralized collagen fibrils in cortical bone under mechanical loading is altered by aging" investigates the morphological and mechanical response of mineralized collagen fibrils due to elastic deformation of the murine tibial cortices in both the control and disused state using animals at different ages. My first suggestion would be to change the title to better reflect the animal study design used in the manuscript. Furthermore, keywords should reflect that this study focuses solely on animal research, as there have been numerous recent investigations of human mineralized collagen fibrils. The abstract requires major restructuring to include data on the number of animals per group and the methodology used to assess mineralized collagen fibrils, among other details. The sentence “In summary, aged bone seems to be more sensitive to disuse” seems too general, and it could be considered as an overstatement since this was not investigated in the study (maybe the data of this study imply that collagen fibrils are more sensitive to disuse in aged animals). Furthermore, the discussion section also devotes excessive attention to the effect of disuse, without acknowledging the covariant effect of aging on bone nano-scale properties. It is unclear why the authors chose to perform EDTA-decalcification of the surface when they aimed to analyze mineralized collagen fibrils. The additional morphological assessment of mineral HA crystals prior to bone surface decalcification could contribute to the study and its findings.

Experimental design

The experimental design needs further elaboration and better phrasing to be easily comprehensible for the readers. Authors should clearly state which groups were used in the study and add a description of these groups one by one. The usage of previous data should be clearly stated for each of the parameters. It is not clear whether all the parameters were the same and whether previous data were used in all of the analyses or not. Also, it would be desirable to be consistent in using the same group names in the text as in the figures. To ensure reader-friendly image representations, I would suggest adding the groups and their details in Figure 2 (number of animals, the use of previous data, etc). Also, a major concern for the current version of the manuscript is the description of AFM analysis. To ensure the reproducibility of the study and its data, parameters for the AFM scanning protocol and a detailed description of image postprocessing and evaluation should be added. It would be very helpful if the authors would consider adding a schematic representation of their approach in AFM assessment. Currently, shame it is just depicted that AFM has been conducted, but the detailed description of the approach would ensure better reproducibility. Further, it is not clear in any way how the regions of interest were selected for this study. Was it a manual, semiautomatic, or automatic protocol? Was it standardized or was it random? How many images per mouse were analyzed? Further, it is not clear why the author decided to do EDTA-decalcification of the surface when they wanted to analyze mineralized collagen fibrils. The morphological assessment of these mineral crystals could contribute to the study and its findings.

Validity of the findings

Study limitations should be clearly stated in the manuscript to provide additional support for the validity of the data in the current literature. I would also suggest that authors add individual data points to the graph bars and boxplots used in figures 3, 4, and 6 to clearly depict all the data generated in the study.

Additional comments

The quality of the English language used in the manuscript needs improvement. The abbreviations should be checked (it should be defined at first mention in the text and used consistently afterwards).

---

## Round 0.2 · accepted · Accept

· Academic Editor

Accept

Dear Dr. Yang,

Both reviewers have found your latest version of the manuscript ready for publication. We are therefore submitting it in its current form for technical preparation for publication.

Best wishes,
Alexander Ereskovsky

Reviewer 1 ·

Basic reporting

Pass after correction to the abstract and methods. After checking the raw data I noticed some AFM images were acquired in contact mode and not just in tapping mode. So line 28 remove Tapping mode and add "or Contact mode imaging" line 176. Ultimately I do not believe the use of two different imaging modes invalidate the data presented.

Experimental design

Pass

Validity of the findings

Pass

Additional comments

All my previous comments were addressed satisfactorily

·

Basic reporting

The manuscript titled "In situ mechanical behavior of mineralized collagen fibrils in murine cortical bone is altered with aging and disuse" investigates the morphological and mechanical response of mineralized collagen fibrils due to elastic deformation of the murine tibial cortices in both the control and disused state, using animals at different ages. The revised manuscript is written in a direct and professional manner, making it clear for a wider scientific audience. The revised version of the methodology provides a clearer understanding of the study design and facilitates study replication. The conclusion of the study is bound to the chosen study design.

Experimental design

The manuscript titled "In situ mechanical behavior of mineralized collagen fibrils in murine cortical bone is altered with aging and disuse" investigates the morphological and mechanical response of mineralized collagen fibrils due to elastic deformation of the murine tibial cortices in both the control and disused state, using animals at different ages. The revised manuscript now provides detailed descriptions of animal groups, with a special emphasis on previously published data. The revised version of the methodology offers a clearer understanding of the study design and facilitates the replication of the study.

Validity of the findings

The revised version of the methodology offers a clearer understanding of the study design and facilitates the replication of the study. The conclusion of the study is bound to the chosen study design.